# A Glycemia-Based Nomogram for Predicting Outcome in Stroke Patients after Endovascular Treatment

**DOI:** 10.3390/brainsci12111576

**Published:** 2022-11-18

**Authors:** Chengfang Liu, Yuqiao Zhang, Xiaohui Li, Yukai Liu, Teng Jiang, Meng Wang, Qiwen Deng, Junshan Zhou

**Affiliations:** Department of Neurology, Nanjing First Hospital, Nanjing Medical University, No. 68 Changle Road, Nanjing 210006, China

**Keywords:** endovascular treatment, glycemia, nomogram, outcome, stroke

## Abstract

**Objective**: Higher fasting glucose is thought to be associated with adverse outcome in patients receiving endovascular treatment (EVT), while the effect of glycosylated hemoglobin (HbA_1c_) on outcome is controversial. We combined fasting blood glucose (FBG) with HbA_1c_ and evaluated their relationship with the three-month functional outcome in patients who underwent EVT. **Methods**: Data from 739 consecutive ischemic stroke patients who underwent EVT from April 2015 to August 2021 were retrospectively reviewed. HbA_1c_ was used to estimate the chronic glucose level according to the following formula: chronic glucose level (mg/dL) = 28.7 × HbA_1c_ (%) − 6.7. Patients were split into two groups in accordance with the three-month modified Rankin Scale (mRS). Univariate and multivariate analyses were utilized to investigate the association of outcome with blood glucose and to identify other predictors of prognosis. **Results**: Patients with poor outcome had significantly higher FBG, chronic glycemia, FBG/chronic glycemic ratio, and difference between FBG and chronic glycemia (Δ_A-C_). FBG, the FBG/chronic glycemic ratio, and Δ_A-C_ remained to be associated with poor outcome after adjustment. We then established a glycemia-based nomogram with a concordance index of 0.841, and it showed favorable clinical utility according to decision curve analysis. **Conclusions**: Glycemia after EVT was connected with the functional outcome and a nomogram based on glycemia may be used to predict prognosis in stroke patients treated with EVT.

## 1. Introduction

Hyperglycemia is usually thought to be associated with worse functional outcome including an increased risk of mortality and symptomatic intracranial hemorrhage (sICH) [1,2,3,4], although another study found no interaction between increased blood glucose and poor prognosis in patients undergoing endovascular treatment (EVT) [5]. These aforementioned studies have focused on the relationship between admission glucose and outcome, while glucose level is dynamically changing and influenced by several factors like stress response and stroke deterioration [6]. Impaired fasting glucose in the first week was found to be associated with poor functional outcome at discharge after EVT [7]. Higher fasting glucose on the next day after EVT is also an independent risk factor for worse three-month outcome [8,9]. Glycosylated hemoglobin (HbA_1c_) is supposed to represent estimated average glucose [10]. Higher HbA_1c_ levels at admission were found to be significantly associated with a reduced likelihood of favorable functional outcome in patients with acute large vessel occlusive stroke and diabetes after EVT [11]. Recently, some researchers proposed an index representing stress hyperglycemia through calculating the ratio of fasting blood glucose (FBG) to HbA_1c_ and found it to be detrimental to the prognosis in patients treated with EVT or thrombolysis [12,13]. Glycemia gap, which means absolute glycemia increase, could also reflect stress-induced hyperglycemia, but its prognostic insight was controversial [14,15]. Limited data are available on the relationship between FBG, HbA_1c_, and prognosis in EVT patients.

Nomograms can generate a possibility of a clinical event individually through gathering personal biological and clinical variables and are extensively used in clinical practice [16]. Several nomograms were proposed to estimate the risk of unfavorable outcome at three months in stroke patients who received EVT recently [17,18,19,20]. Age, admission, the National Institute of Health Stroke Scale (NIHSS) score, collateral circulation, and the postoperative Modified Thrombolysis in Cerebral Infarction (mTICI) scale were commonly considered as independent predictors [17,18,19,20]. However, the forecast performance of these models was discrepant, and few studies investigated the effect of glycemia on functional outcome after EVT. In the present study, we combined FBG with HbA_1c_ to evaluate their effect on clinical outcome and established a nomogram to predict prognosis at three months for acute ischemic stroke patients treated with EVT.

## 2. Methods

This is a single-center, retrospective study composed of consecutive acute ischemic stroke patients receiving EVT in Nanjing First Hospital from April 2015 to August 2021. Patients without information on fasting glucose after EVT, HbA_1c_, or functional outcome at three months were excluded. Data regarding demographics, medical history, laboratory examination, NIHSS score at admission, stroke characteristics, collateral circulation, details of procedure, such as recanalization status, and three-month modified Rankin Scale (mRS) were collected. Collateral circulation was graded according to the American Society of Interventional and Therapeutic Neuroradiology/Society of Interventional Radiology (ASITN/SIR). Recanalization status was assessed by the mTICI scale. Symptomatic intracranial hemorrhage (sICH) was defined as any intracranial hemorrhage with NIHSS increase ≥ 4 points from baseline within 24 h [21]. All therapeutic regimen followed the current guidelines [22] and the analysis was approved by the Ethics Committee of Nanjing First Hospital, Nanjing Medical University.

Fasting blood glucose and HbA_1c_ were tested utilizing fasting blood samples which were drawn in the next morning after EVT. Average chronic glycemia was estimated by the following formula: chronic glucose levels (mg/dl) = 28.7 × HbA_1c_ (%) − 46.7 [10]. The above two values were then used to calculate the ratio (A/C) and difference between FBG and chronic glycemia (Δ_A-C_). Patients were classified into two groups according to mRS at three months. A good functional outcome was defined as mRS ≤ 2 and poor outcome was defined as mRS > 2 at three months.

Continuous data are expressed as mean ± standard deviation or median with quartiles, while categorical data are presented as numbers with percentages. Quantitative data were compared with Student’s *t*-test or the Mann–Whitney U test and qualitative data were analyzed by the chi-squared test. Multivariable logistic regression analysis was applied to identify the independent risk factors of poor outcome. Receiver operating characteristic (ROC) curves and the areas under the ROC curves were counted to appraise the predictive power of these blood glucose factors. Based on the above predictors, a nomogram model for three-month functional outcome prediction was established. The discrimination of the nomogram was evaluated by the concordance index (C-index). The calibration capacity was assessed by calibration curve to compare the observed against estimated risk of poor outcome after EVT. The clinical usefulness was measured by decision curve analysis (DCA) to investigate whether nomogram-assisted decisions would improve outcome. Statistical analyses were performed using Statistical Package for the Social Sciences version 20.0 (SPSS Inc., Chicago, IL, USA) and R software version 3.5.2 (Institute for Statistics and Mathematics, Vienna, Austria). A two-tailed *p* value < 0.05 was regarded as statistically significant.

## 3. Results

Among 788 ischemic stroke patients treated with EVT in our center, 48 patients without HbA_1c_ or FBG after EVT and one patient without three-month mRS were excluded. As a result, 739 patients were included in the final analysis (Figure 1). The average age of the patients was 70.0 ± 12.2 years and 471 (63.7%) were men. The mean FBG was 128 ± 45 mg/dl, and HbA_1c_ was 6.3 ± 1.4. Table 1 summarizes the baseline characteristics of the patients according to three-month functional outcome. Patients with good functional outcome were significantly younger and had a higher proportion of males than those with poor outcome. Diabetes, atrial fibrillation, and prior stroke were less prevalent among patients with good functional outcome. They also had lower FBG, average chronic blood glucose, NIHSS score, door-to-first recanalization time and number of devices passed. Patients with good functional outcome had a higher prevalence of large artery atherosclerosis, better collateral circulation, and achieved a higher rate of successful recanalization.

Table 2 shows that FBG (OR: 1.012, 95% CI: 1.006–1.018, *p* < 0.001), A/C glycemic ratio (OR: 4.783, 95% CI: 2.183–10.478, *p* < 0.001) and Δ_A-C_ (OR: 1.007, 95% CI: 1.002–1.011, *p* = 0.006) remained to be associated with poor outcome after multivariable adjustment. ROC analyses showed that the above three blood glucose levels exhibited greater accuracy of predicting poor prognosis than average chronic glycemia (Appendix A). As shown in Table 1, the proportion of diabetes was discrepant in two groups, a subgroup analysis was performed according to the presence of diabetes (Table 3). In patients without diabetes, FBG, A/C glycemic ratio, and Δ_A-C_ were significantly associated with an increased risk of poor outcome at the three-month mark, while the correlations were not significant in patients with diabetes. We also documented three-month mortality and sICH as outcomes and found that FBG, A/C glycemic ratio, Δ_A-C_ remained to be the independent risk factors (Appendix A). Besides, age, baseline serum creatinine, NIHSS score, ASITN/SIR, and number of devices passed were also predictors of three-month mortality. In the multivariable logistic regression analysis, ASITN/SIR and number of devices passed were connected with odds of sICH.

Abbreviations: OR, odds ratio; FBG, fasting blood glucose; A/C, FBG/chronic; Δ_A-C_, the difference between FBG and chronic glycemia.

To further precisely predict clinical outcome, we established a nomogram for forecasting the prognosis at three months based on glycemia and significant variables. As shown in Figure 2, each factor had a corresponding score from 0 to 100 and then we could calculate the total score to predict the risk of poor functional outcome in patients who underwent EVT. The C-index of the nomogram was 0.841, whereas its value was 0.811 when the aforementioned three blood glucose factors were not considered. We could see favorable consistency between the predicted and actual risk of poor prognosis from the calibration curve (Figure 3). Decision curve analysis showed that it will be of much clinical value to utilize the nomogram with glycemia or without glycemia in a wild range (Figure 4). Because the connection between aforesaid glycemia factors and outcome in non-diabetic patients was remarkable, we also provided a separate nomogram based on these independent risk factors for nondiabetics (Appendix A).

## 4. Discussion

Our study showed that FBG and chronic glycemia were associated with functional outcome in patients with large vessel occlusion who underwent EVT. We present a nomogram model based on age, admission NIHSS, ASITN/SIR collateral grading, number of devices passed, postoperative mTICI scale, FBG, the FBG/chronic glycemic ratio, and Δ_A-C_ to predict the probability of poor outcome. This would provide clinicians with a rapid and precise tool for prognosis prediction.

Hyperglycemia is commonly defined as blood glucose level > 140mg/dL and approximately one-third of stroke patients with large vessel occlusion had admission hyperglycemia [1,3,23]. Previous studies have revealed that patients with admission hyperglycemia are at an increased risk of unfavorable functional outcome, sICH, and mortality after EVT [1,4,23]. However, Osei et al. did not find the interaction of admission glucose or hyperglycemia with mRS at 90 days [5]. Yong et al. observed that hyperglycemia at 24 h was related to an enlarged chance of parenchymal hemorrhage within 7 days, dependent at three months, and mortality within 90 days in acute ischemic hemispheric stroke patients [6]. On the contrary, patients with hyperglycemia at baseline and declined serum glucose at 24 h did not have an increased risk of worse outcome [6]. Li et al. also found that postoperative glucose measured within 24 h after EVT instead of glucose at admission was independently connected with sICH [24]. Blood glucose level is dynamically changing, which is not only a stress response, but also affected by stroke severity [6]. It implies that we should also pay attention to blood glucose after EVT rather than concentrate on admission glucose. Nevertheless, random blood glucose is susceptible to a variety of factors, like intake of food rich in sugar and medicine like hypoglycemic agents [9] et al. Fasting blood glucose is less affected by food and drugs and almost every patient was tested after admission. FBG could partly reflect stress response resulting from the interaction of hormones with concomitant insulin resistance during acute diseases [9,25]. To some extent, it could reflect the influence of EVT on blood glucose. Only a few studies indicate that increased fasting blood glucose rises the risk of adverse functional outcome after EVT [7,8,9,25]. A retrospective cohort study found that there was a shift towards worse functional outcome in patients with increased fasting glucose levels, and impaired fasting glucose (glucose level above 5.5 mmol/L) was associated with poor functional outcome or death and with sICH during hospitalization [7]. However, the fasting glucose is measured in the first week after EVT. Wnuk [8] and Yuan [9] found that higher fasting glucose levels detected on the next day after EVT increased the risk of unfavorable outcome at three months. Nevertheless, they had a limited sample size with only 181 patients [8] or the association was only found in patients aged ≥60 years [9]. In the present study, we found that patients with unfavorable outcome had higher fasting blood glucose (138 ± 48 mg/dL) than patients with favorable outcome (112 ± 36 mg/dL). After adjustment for potential confounders, it is still an independent predictor of the three-month outcome.

However, an isolated blood glucose index such as fasting concentrations cannot distinguish between chronic hyperglycemia and a physiologic stress response to stroke [26]. Thus, we combined FBG with HbA_1c_ to evaluate their effect on clinical outcome. HbA_1c_ can estimate average chronic glucose value [10] and reflect pre-stroke glucose control [11]. A prospective cohort study demonstrated that patients with high HbA1c levels had half the chance of achieving a favorable outcome and four times higher risk of mortality than those who did not [27]. Diprose et al. discovered that there were 24% lower odds of functional independence, 26% higher odds of mortality, and 33% higher odds of sICH for every 10 mmol/mol increase in HbA1c [28]. Chang et al. found that HbA1c levels with a range of 7.0–7.1% at admission were significantly related to decreased odds of favorable outcome in diabetics after undergoing EVT [11]. High HbA_1c_ levels and favorable functional outcome were still significantly negatively connected in patients with posterior circulation large vessel occlusion who received EVT [29]. However, Li et al. did not find that HbA_1c_ could predict poor outcome [30]. Our study showed that patients with poor outcome had higher HbA_1c_ and A/C glycemic ratio than those with good outcome. In the multivariable logistic regression analysis, the association of chronic glycemia with unfavorable outcome was insignificant, while the ratio of FBG to chronic glycemia remained an independent predictor of poor outcome. The finding is supported by Chen et al., who found that increased stress hyperglycemia, counted by the ratio of fasting glucose to estimated average glucose concentration, was a strong forecaster of poor clinical outcome, while there was no statistical difference in HbA_1c_ between patients with or without favorable outcome [26]. Unlike other studies that investigated single glucose level like FBG, HbA_1c_, or stress hyperglycemia ratio, we not only combined the above indicators, but also evaluated the difference of fasting glucose and chronic glycemia. We found that FBG, FBG/chronic glycemic ratio, and Δ_A-C_ all exhibited moderate predictive value for poor functional outcome. They were also significant independent predictors of sICH and three-month mortality.

Diabetes is not merely a risk factor of cerebrovascular accident and may affect prognosis. A meta-analysis discovered that patients with a history of diabetes were associated with significantly lower odds of functional independence at three months after EVT compared with those without diabetes [31]. Wnuk et al. found that fasting hyperglycemia (glucose level above 5.5 mmol/L) was linked to poor neurological outcome in patients without diabetes but not with diabetes [8]. Chen also found the relationship between stress hyperglycemia ratio > 0.96 and poor outcome remained significant after adjusting for potential covariates in nondiabetics, but the connection was insignificant in diabetics [26]. In our study, patients with poor prognosis also had higher a proportion of diabetes (34.4%), while in the subgroup analysis we did not find evidence that FBG, FBG/chronic glycemic ratio, and Δ_A-C_ elevation increase the risk of poor prognosis at the three-month point in diabetic patients receiving EVT. Nevertheless, the associations between the above indexes and functional outcome in nondiabetic patients were strong. This may be attributed to patients without diabetes being more susceptible to glucose fluctuations, while diabetics had a high tolerance. This indicated that acute blood glucose increase may be a significant predisposing factor in poor functional outcome. Therefore, we also established a nomogram for nondiabetics, which showed better discrimination with C-index of 0.855 than the nomogram for all patients. The potential mechanisms of glycemia increase on poor functional outcome may attributed to several aspects, such us altered mitochondrial function, disrupted blood–brain barrier, increased oxidative stress response, impaired cerebrovascular reactivity, and exacerbated ischemic brain injury, among others [7,9,25,32,33].

Nomogram is a simple and intuitive tool for forecasting prognosis and assessing risk which could guide clinical treatment decisions. Several studies have proposed nomograms to predict outcome after thrombectomy [17,18,19,20]. Age, gender, pre-stroke mRS score, NIHSS score, collateral circulation, postoperative TICI score, and several other factors could be used to build a nomogram for predicting the possibility of poor outcome in patients receiving EVT [17,18,19,20]. However, none of these studies explored the establishment of a prognostic model based on blood glucose. In the present study, we found that FBG, FBG/chronic glycemic ratio, and Δ_A-C_ played an important role in functional outcome of ischemic stroke patients. The likelihood of achieving good functional outcome decreased as blood glucose level increased. We also discovered several other predictive factors like age, admission NIHSS, ASITN/SIR collateral grading, number of devices passed, and postoperative mTICI scale, which were reported to be associated with prognosis. Baseline NIHSS reflected stroke severity and patients with high NIHSS scores may have sequela even after recanalization. EVT is the vital treatment for acute large vessel occlusion and successful recanalization is a powerful forecasting factor of good outcome [19,34]. Our study confirmed that patients with higher recanalization degree had higher likelihood of better prognosis. The number of devices passed was found to be negatively associated with favorable outcome. Linfante et al. investigated predictors of poor prognosis after successful recanalization and showed that ≥3 passes was significantly related to a poor three-month outcome [34]. They explained that a higher number of devices passed was connected with longer recanalization time. Seker et al. also revealed that the probability of obtaining poor prognosis increased with every thrombectomy maneuver and the chance of achieving a good prognosis was the highest when patients were recanalized within two maneuvers [35]. The connection between number of devices passed and outcome was unknown. It may also be attributed to progressive infarction during prolonged procedure time and barely removed platelet-rich thick clots, which led to unsuccessful recanalization [35]. Based on these factors, we established a nomogram with C-index of 0.841 which showed good discrimination. It means that we could discern a patient with poor functional outcome from a patient without poor prognosis 84.1% of the time. The calibration plot suggested the nomogram prediction is close to actual risk. Moreover, the clinical applicability of the nomogram was demonstrated by DCA. In a wide range of threshold probabilities, it would be of net benefit to patients to make treatment decisions in line with a prediction model compared to the decisions made by assuming that either all or no patients have poor prognosis. By gathering the necessary information of the above eight predictors pre- and post-treatment, we could utilize the nomogram to acquire a visualized and reliable prognosis prediction. It may also provide some strategies for blood glucose management after EVT. Up to now, the benefits of intensive glucose treatment on favorable outcome or death were not identified in patients treated with EVT [36,37]. Instead, intensive glucose treatment may lead to severe hypoglycemia. However, the above two studies both had small sample sizes. Our findings suggested that it may be beneficial to avoiding adverse outcome by maintaining FBG around 130 mg/dL.

Our study has several limitations. First, this is a retrospective study performed in a single center. Second, we only evaluated serum glucose levels on the next day after EVT, so we did not know the blood glucose fluctuation during hospitalization, which may also have influenced prognosis. Third, the information on utilization of hypoglycemic agents both before stroke and in hospital was lacking. Forth, the nomogram was developed in a limited sample size and needs to be validated in a larger external population.

## 5. Conclusions

Higher FBG, FBG/chronic glycemic ratio and Δ_A-C_ were significantly associated with functional outcome in patients treated with EVT. A nomogram composed of age, admission NIHSS, ASITN/SIR collateral grading, number of devices passed, postoperative mTICI scale, and above glycemia factors may predict the risk of adverse outcome. Future studies are warranted to validate the nomogram and investigate the benefits of hypoglycemic therapy.

## Figures and Tables

**Figure 1 brainsci-12-01576-f001:**
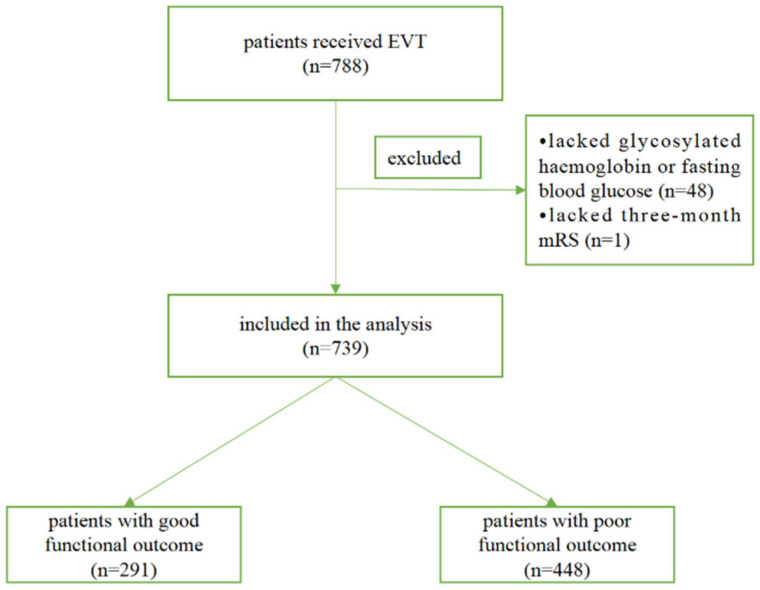
Study flow diagram. Good functional outcome was defined as mRS ≤ 2 and poor functional outcome was defined as mRS > 2 at three months.

**Figure 2 brainsci-12-01576-f002:**
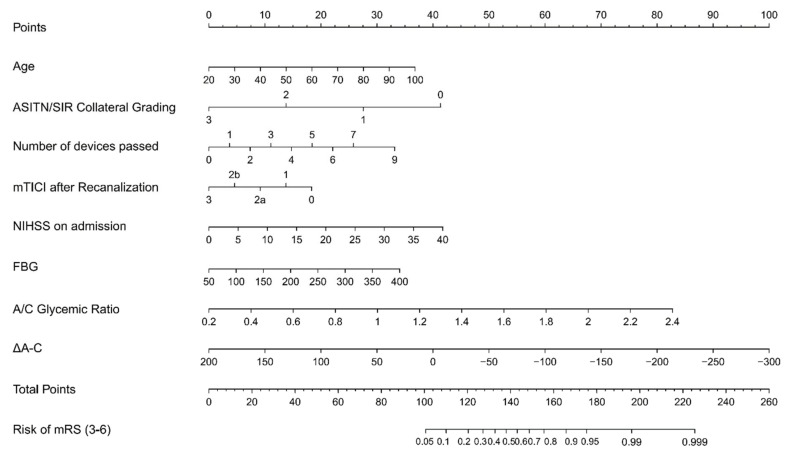
Nomogram for predicting the probability of three-month poor outcome in ischemic stroke patients receiving EVT. We drew a line perpendicular to the point scale axis and added the points for all variables. Then, we could estimate the risk of poor outcome through the total points. ASITN/SIR, American Society of Interventional and Therapeutic Neuroradiology/Society of Interventional Radiology; mTICI, Modified Thrombolysis in Cerebral Infarction; NIHSS, National Institutes of Health Stroke Scale; FBG, fasting blood glucose; A/C, FBG/chronic; Δ_A-C_, the difference between FBG and chronic glycemia; mRS, modified Rankin Scale.

**Figure 3 brainsci-12-01576-f003:**
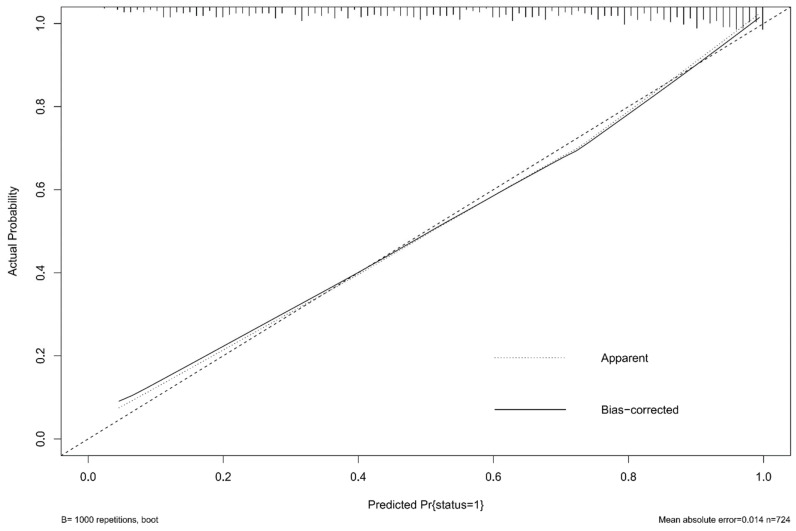
Calibration curve of the nomogram for predicting outcome.

**Figure 4 brainsci-12-01576-f004:**
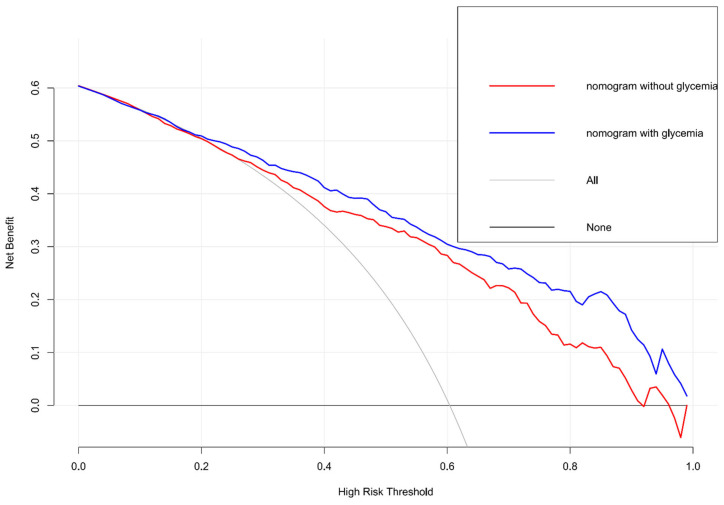
Decision curve analysis for the nomogram.

**Table 1 brainsci-12-01576-t001:** Baseline characteristics of patients.

	Total (739)	mRS 0–2 (*n* = 291)	mRS 3–6 (*n* = 448)	*p*
Age (years), mean ± SD	70.0 ± 12.2	65.5 ± 12.7	72.9 ± 10.9	<0.001
Sex, male, *n* (%)	471 (63.7%)	208 (71.5%)	263 (58.7%)	<0.001
Medical history, *n* (%)				
Hypertension	547 (74.0%)	205 (70.4%)	342 (76.3%)	0.074
Diabetes	234 (31.7%)	80 (27.5%)	154 (34.4%)	0.049
Atrial fibrillation	334 (45.2%)	99 (34.0%)	235 (52.5%)	<0.001
Prior stroke	152 (20.6%)	49 (17.0%)	103 (23.0%)	0.048
Laboratory examination, mean ± SD				
FBG, mg/dL	128 ± 45	112 ± 36	138 ± 48	<0.001
HbA_1c_, %	6.3 ± 1.4	6.2 ± 1.2	6.4 ± 1.5	0.013
Average chronic glycemia, mg/dL	135 ± 40	130 ± 35	138 ± 43	0.013
A/C glycemic ratio	0.97 ± 0.28	0.88 ± 0.22	1.03 ± 0.30	<0.001
Δ_A-C_, mg/dL	−6 ± 43	−18 ± 36	1 ± 46	<0.001
Serum creatinine, μmol/L	77.2 ± 32.6	75 ± 33	78 ± 32	0.245
Total cholesterol, mg/dL	76 ± 21	77 ± 20	76 ± 22	0.489
Triglycerides, mg/dL	23 ± 16	23 ± 14	23 ± 17	0.932
HDL, mg/dL	20 ± 6	20 ± 5	20 ± 7	0.077
LDL, mg/dL	46 ± 17	47 ± 17	46 ± 17	0.275
Baseline NIHSS score, median (IQR)	14 (10–18)	12 (7–16)	16 (12–20)	<0.001
Infarct circulation, *n* (%)				0.464
Anterior	626 (84.7%)	243 (83.5%)	383 (85.5%)	
Posterior	113 (15.3%)	48 (16.5%)	65 (14.5%)	
Stroke subtypes, *n* (%)				<0.001
LAA	332 (44.9%)	152 (52.2%)	180 (40.2%)	
CE	349 (47.2%)	106 (36.4%)	243 (54.2%)	
SOE	22 (3.0%)	17 (5.8%)	5 (1.1%)	
SUE	36 (4.9%)	16 (5.5%)	20 (4.5%)	
ASITN/SIR, median (IQR)	2 (1-2)	2(2-2)	1 (1-2)	<0.001
Interval time, min, median (IQR)				
Onset to door	175 (86–308)	175 (85–305)	175 (81–300)	0.924
Door to groin puncture	107 (80–140)	108 (80–138)	104 (78–140)	0.283
Door to first recanalization	184 (149–228)	170 (144–214)	190 (150–230)	0.003
Intravenous thrombolysis, *n* (%)	309 (41.8%)	132 (45.3%)	177 (39.5%)	0.119
Number of devices passed, median (IQR)	2 (1-3)	1 (1-2)	2 (1-3)	<0.001
mTICI score, *n* (%)				<0.001
2b-3	644(87.1%)	276(94.8%)	368(82.1%)	
0-2a	95(12.9%)	15(5.2%)	80(17.9%)	

Abbreviations: mRS, modified Rankin Scale; SD, standard deviation; FBG, fasting blood glucose; HbA_1c_, glycosylated hemoglobin; A/C, FBG/chronic; Δ_A-C_, the difference between FBG and chronic glycemia; HDL, high density lipoprotein; LDL, low-density lipoprotein; NIHSS, National Institutes of Health Stroke Scale; IQR, interquartile range; LAA, large artery atherosclerosis; CE, cardiac embolism; SOE, stroke of other determined etiology; SUE, stroke of undetermined etiology; ASITN/SIR, American Society of Interventional and Therapeutic Neuroradiology/Society of Interventional Radiology; mTICI, Modified Thrombolysis in Cerebral Infarction.

**Table 2 brainsci-12-01576-t002:** Unadjusted and adjusted ORs of glycemia and other baseline characteristics for poor functional outcome.

	Crude OR (95% CI)	*p*	Adjusted OR (95% CI)	*p*
FBG	1.017 (1.012–1.022)	<0.001	1.012 (1.006–1.018)	<0.001
Chronic glycemia	1.005 (1.001–1.009)	0.019	1.005 (0.999–1.011)	0.122
A/C glycemic ratio	10.720 (5.559–20.671)	<0.001	4.783 (2.183–10.478)	<0.001
Δ_A-C_	1.011 (1.007–1.015)	<0.001	1.007 (1.002–1.011)	0.006
Age	1.055 (1.040–1.069)	<0.001	1.041 (1.022–1.060)	<0.001
Sex	0.567 (0.413–0.778)	<0.001	0.792 (0.533–1.177)	0.249
Hypertension	1.354 (0.970–1.888)	0.075		
Diabetes	1.382 (1.000–1.908)	0.050		
Atrial fibrillation	2.140 (1.577–2.904)	<0.001	0.823 (0.538–1.259)	0.369
Prior stroke	1.462 (1.002–2.134)	0.049	0.970 (0.609–1.544)	0.896
HDL	1.024 (0.997–1.051)	0.079		
Baseline NIHSS score	1.122 (1.093–1.151)	<0.001	1.098 (1.067–1.130)	<0.001
Stroke subtypes	0.995 (0.855–1.157)	0.946		
ASITN/SIR	0.253 (0.189–0.339)	<0.001	0.289 (0.208–0.402)	<0.001
Door to first recanalization	1.001 (0.999–1.003)	0.213		
Number of devices passed	1.507 (1.317–1.724)	<0.001	1.352 (1.147–1.594)	<0.001
mTICI score	0.250 (0.141–0.443)	<0.001	0.334 (0.169–0.660)	0.002

Adjusted for age, sex, hypertension, diabetes, atrial fibrillation, prior stroke, HDL, baseline NIHSS score, stroke subtypes, ASITN/SIR, door to first recanalization time, number of devices passed and mTICI score. Abbreviations: OR, odds ratio; FBG, fasting blood glucose; A/C, FBG/chronic; Δ_A-C_, the difference between FBG and chronic glycemia.

**Table 3 brainsci-12-01576-t003:** Subgroup analysis of glycemia for poor functional outcome with patients with or without diabetes.

	Patients with Diabetes (*n* = 234)				Patients without Diabetes (*n* = 505)				P-Interaction
	Crude OR (95% CI)	P	Adjusted OR (95% CI)	P	Crude OR (95% CI)	P	Adjusted OR (95% CI)	P	Diabetes and glycemia
FBG	1.008 (1.002–1.014)	0.004	1.006 (0.999–1.012)	0.091	1.035 (1.026–1.044)	<0.001	1.025 (1.014–1.035)	<0.001	0.004
Chronic glycemia	1.002 (0.996–1.008)	0.495			1.011 (1.000–1.023)	0.056			0.039
A/C glycemic ratio	3.092 (1.299–7.359)	0.011	1.656 (0.590–4.649)	0.339	46.832 (16.923–129.602)	<0.001	15.735 (4.588–53.969)	<0.001	0.005
Δ_A-C_	1.004 (1.000–1.009)	0.046	1.001 (0.996–1.006)	0.753	1.033 (1.024–1.041)	<0.001	1.024 (1.014–1.035)	<0.001	0.123

Adjusted for age, sex, hypertension, atrial fibrillation, prior stroke, HDL, baseline NIHSS score, stroke subtypes, ASITN/SIR, door to first recanalization time, number of devices passed and mTICI score.

## Data Availability

The datasets generated during and/or analyzed during the current study are available from the corresponding author on reasonable request.

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
