# Peer review of "A Glycemia-Based Nomogram for Predicting Outcome in Stroke Patients after Endovascular Treatment"

_brainsci, 2022, doi:10.3390/brainsci12111576_

Round 1

Reviewer 1 Report

In this paper, the authors carry out a nomogram that associates glycaemia with functional outcome after an acute ischemic stroke. I think this topic is interesting and very up-to-date. The introduction is correct, the methods are sound, and the results and conclusions are logical and well presented. The main limitation, as the authors already point out, is the absence of glycaemia prior to mechanical thrombectomy. Otherwise, I think the work is of good quality.

Reviewer 2 Report

Reviewer’s Comments on the paper “A glycemia-based nomogram for predicting outcome in stroke patients after endovascular treatment.”

The present work represents a well-conducted, retrospective analysis of a large cohort of stroke patients who underwent EVT, providing clinicians with a useful prognostic tool able to predict three months patients’ functional outcomes based on acute and chronic blood glucose levels after admission. Although there is no evidence of a beneficial effect of aggressive blood glucose lowering therapy in AIS patients, the nomogram proposed by the authors might be useful for the selection of patients for future clinical trials on this topic. However,  the study requires the following adjustments:

1.    Both in the introduction and in the discussion section, you reported evidence of an association between acute hyperglycemia and mortality and sICH in AIS patients undergoing revascularization treatment. Although the mRS 3-month outcome alone might be already quite informative of patients’ functional independence, I think that adding three months mortality rate and risk of sICH to the analysis would further improve the quality of the present work, aligning it to other reports available on the topic.

2.    Patients with three months poor functional outcomes had a most severe stroke in terms of NIHSS at onset, mTICI score, and number of devices passed, which were significant predictors in the multivariate analysis, even though with a lower OR compared to A/C glycaemic ratio. Please comment on this finding in the discussion section.

3.    I found very interesting the subgroup analysis between diabetic and non-diabetic patients, which showed that hyperglycemia is able to predict a worse functional outcome only in non-diabetic patients. Please provide a separate nomogram for this subgroup only and further comment on this in the discussion section.

Reviewer 3 Report

The aim of the study is to evaluate the effect of fasting blood glucose and HbA1c on clinical outcome, and to establish a nomogram to predict prognosis at 3 months for acute ischemic stroke patients that underwent EVT. The strengths of the study include inclusion of a large dataset of 739 patients, and the use of a nomogram that incorporates blood glucose levels.

General concept comments

Defining the fasting blood glucose (FBG) the morning after EVT as acute glycemia or acute glucose level is misleading to most readers, as it does not take into consideration the blood glucose level of patients on admission to the acute stage after EVT, it is a single measurement only. It is better to use the term “FBG after EVT” rather than “acute glycemia”.

Include a section to justify why blood glucose levels on admission were not used in addition to the fasting blood glucose after EVT. Considering that admission blood glucose levels have been associated with outcome in a number of other studies, the addition of this to the nomogram would be beneficial if the data is available.

Since the correlations were not significant in patients with diabetes (Lines 133-135), please justify why diabetes was not included in the nomogram. Also mention the usefulness of the nomogram in patients with diabetes and those taking antidiabetic medications or insulin. 

Specific comments

Unify the terminology across the paper either mention as glycemia or glycaemia.

Figure 1 – Define good and poor functional outcome in the legends.

Figure 2 – Image is not clear. Provide a higher quality image and more description under figure legends.

Figure 3 – Provide a higher quality image. The text is hazy and unclear.  
